# His unemployment, her response, and the moderating role of welfare policies in European countries

**Anna Matysiak**[1]*, **Anna Kurowska**[2], **Alina Maria Pavelea**[1]

**1** Faculty of Economic Sciences, University of Warsaw, Warsaw, Poland, **2** Faculty of Political Science and International Studies, University of Warsaw, Warsaw, Poland

* annamatysiak@uw.edu.pl

## Abstract

Recent changes in labour markets have increased employment instability. Under these conditions, in male breadwinner families women might increase their labour supply when their male partners become unemployed. Previous studies have extensively investigated the role played by household and individual characteristics in explaining such increases in the labour supply of women. However, studies which examine the moderating role of specific welfare policies are missing. Our study contributes to the literature by investigating the moderating effect of childcare and tax-benefit policies for the labour supply response of women following the unemployment of their partner. We focus on a sample of 24 EU member states and the UK, during the period 2009-2019, combining longitudinal microdata from EU-SILC with country-period specific policy indicators generated with the use of the tax-benefit simulation model EUROMOD, UKMOD and country-period specific indicators of childcare use.

## Introduction

Despite a massive increase in women's education and employment, across all EU Member States women are still less often present in employment compared to men [1]. They also work part-time more often and earn lower wages [2]. These differences usually become more pronounced around the time of the first birth [3, 4], when women reduce their economic activity in order to take care of their children. Even though men have increased their contribution to housework and have taken an increasing proportion (albeit still small) of parental leave [5], women continue to be primary caregivers of young children [6]. They usually take longer career breaks from employment after birth or reduce their economic activity. Furthermore, the proportion of women who do not work or look for a job remains higher than that of men [7]. As a result traditional and modified male breadwinner families, in which women are inactive or working a reduced number of hours, are still prevalent in Europe [8].

At the same time, employment is less stable than it once was. Growing competition in the labour markets, caused by ongoing globalization, digitalisation, and deregulation reforms undertaken in order to increase the flexibility of employment relations, have resulted in greater employment instability [9]. Industrial relationships typical for the Fordist production model,

**Data Availability Statement:** All relevant data from this study will be made available upon study completion. The microdata intended to be used for the present registered research protocol need to be requested according to the following procedure: https://ec.europa.eu/eurostat/documents/203647/771732/How_to_apply_for_microdata_access.pdf. We are not allowed to share these data with anyone

due to the regulations imposed by the European Commision on the use of the Eurostat microdata. We will however deposit our codes in the Zenodo repository with instructions on how to access the data.

**Funding:** This project has received funding from the European Union's Horizon Europe research executive agency (REA) under Grant Agreement No.101060410. Views and opinions expressed are however those of the author(s) onlyand do not necessarily reflect those of the European Union or [name of the granting authority]. Neitherthe European Union nor the granting authority can be held responsible for them The funders had no role in study design, data collection and analysis, decision to publish, or preparation of the manuscript.

**Competing interests:** The authors have declared that no competing interests exist.

based on mutual long-term commitment and employers' responsibility for employment stability, have been replaced by more flexible worker-firm relations which imply more occupational changes, more unstable and temporary job contracts and higher rates of job loss [10, 11]. In these new circumstances, career breaks, due to either involuntary job loss, the end of a temporary contract or job mobility, are much more likely. These changes can be especially detrimental to traditional and modified male breadwinner families, given that they predominantly rely on one income.

Consequently, men's unemployment has important consequences for these types of families and may trigger a response from women. First and foremost, it means an essential loss of income for the family, thus encouraging the female partner to increase her working hours or start searching for a job if she is inactive. If the man takes over some or all of the housework and childcare, men's unemployment may also free women to invest more time and effort in their occupational career [12]. In any case, a man becoming unemployed should theoretically imply an increase in the labour supply of his female partner, an effect called 'added-worker effect' [13]. This reaction may, however, be further moderated by the country's context and its welfare policies. In particular, it may be weaker in countries with more generous safety nets, such as unemployment benefits and social allowances (e.g. Nordic countries) than in countries where families must rely largely on their own resources (e.g. many Southern or Central and Eastern European countries). Likewise, women, especially mothers of young children, may be more likely to increase their labour supply in countries with high public childcare availability. Public childcare, together with the male partner, can allow women to be less involved in the provision of care and to pursue economic activity on a larger scale.

Women's labour supply response to a partner's becoming unemployed has been the subject of many empirical investigations. Some studies have indeed found support for the added-worker effect, though findings vary strongly by country [14–16]. The cross-country variation was often attributed to differences in welfare state regimes [16, 17]. However, studies examining the role of specific welfare policies have yet to be conducted.

This study primarily aims to fill this research gap, by answering the research question: "How do welfare policies moderate women's labour supply increase following their partner's unemployment?". We thus investigate how specific welfare state policies moderate the labour supply response of women in Europe, when their partner becomes unemployed. In particular, we examine the role of the generosity and design of social benefits and tax systems in incentivizing or disincentivizing women's response. We also investigate the enabling role of childcare provision in this process. One contribution of our study is that we do not restrict the analyses only to male breadwinner couples (male employed, female inactive), as the majority of studies have done [18–21], but also consider couples where the woman works part-time and might increase her working hours. This is highly relevant for the European countries, many of which are characterized by relatively high levels of women's labour force participation, albeit on a part-time basis [22]. Our focus is on 24 EU Member States over the period 2009–2019. Our study is based on data from multiple sources: EU-SILC longitudinal and cross-sectional datasets, EUROMOD, UKMOD, European Values Survey and Eurostat database. We use the EU-SILC longitudinal data to model women's reactions to their partner's unemployment at the individual level; the EU-SILC cross-sectional data to compute the childcare coverage by age categories; EUROMOD to assess how the generosity and design of the system of social transfers and taxes affects a woman's reaction to her partner's unemployment; Eurostat and the European Values Survey to retrieve country level control variables. The policy indicators are linked to the EU-SILC data in a multi-level framework so that the moderating effects of specific welfare policies on women's labour supply response to their partners' unemployment can be estimated. In this study we focus on different sex couples. Unfortunately, the data we

use does not allow us to investigate how partners in same-sex couples react to the unemployment of their partner.

## Theoretical background and hypotheses

Multiple theoretical frameworks have been proposed to predict women's reaction to the unemployment of their male partner. The economic concept of the 'added-worker effect' presupposes that when the male partner becomes unemployed, the female partner will either enter the labour market if she has been inactive, or will increase her working hours [13]. The reasons for this increase are twofold. First, a woman may increase her economic activity in order to compensate for the drop in household income and to avoid economic hardship. Second, she may also do so if the relative value of her non-market time is reduced, assuming that her unemployed partner takes up the unpaid work previously done by her [13]. A similar explanation, the 'time-availability hypothesis', was put forward in the sociological literature [12]. It presupposes that a partner who is less involved in paid work is more available to do housework and childcare, allowing the other partner to spend more time on economic activity. This hypothesis acknowledges the time constraints women face because of their domestic and childcare responsibilities [23], which act as a barrier to their labour-force participation [24]. In any case, be it economic or time-related, these considerations lead to our first hypothesis:

*H1. Women will increase their labour market supply in response to their male partner's unemployment*

This hypothesis is founded on the assumption that the male partner will take over the household and childcare responsibilities after becoming unemployed. There is evidence, however, that unemployed men do not always substitute their female partners in fulfilling their domestic and care responsibilities [25, 26] and that they have a tendency to take over only specific and usually non-routine types of childcare/housework tasks [12]. Furthermore, women may even increase the time they spend on housework and/or childcare after their male partner becomes unemployed. Such a reaction, known as 'doing gender' [27] or 'gender deviance neutralization' [28], aims at reinstalling the traditional division of labour where there is a risk it can be violated [29]. These two mechanisms, man's failure to substitute a woman in her domestic responsibilities and doing gender, may hinder women from increasing their labour supply after their male partner becomes unemployed. This may be particularly the case when there are young children at home, as fathers tend to be more involved in taking care of older children than infants [30]. This leads us to the formulation of the second hypothesis:

*H2. Women's labour supply response will be weaker among mothers, especially those who have small children*

For mothers, whose male partners may be less likely to take over childcare responsibilities, widely available and affordable childcare services may serve as an alternative opportunity to outsource childcare and increase labour supply. This reasoning is consistent with studies that find positive effects of childcare supply on mother's labour force participation [31, 32]. Thus, we expect that:

*H3. The labour supply response of mothers will be stronger in countries with higher availability of formal childcare*

Beyond childcare policies, tax-benefit systems may also affect women's labour market response to the unemployment of their male partners. Generous social transfers (e.g. unemployment benefit, social assistance, family benefits) buffer the income loss and reduce the

financial incentive for women to work [33, 34]. Consequently, the more generous the financial transfers, the less likely women are to increase their labour supply in response to the unemployment of their partner. This will be the case, however, provided that the main reason women increase their labour supply following the unemployment of their partners is to compensate for lost income. The effect of the generosity of social transfers has not yet been considered in studies investigating how women respond to their partner's unemployment. Based on these considerations, we formulated the fourth hypothesis:

*H4. The labour supply response of women will be weaker in countries where the tax-benefit system is more generous in replacing the income loss caused by the unemployment of the male partner*

Finally, a woman's reaction may not only depend on the generosity of the social transfers but also on the eligibility rules and in particular on the amount of income which is "taxed away" [35] relative to the earnings she gains after she enters the labour market or expands her working hours. These income losses encompass social benefits which are withdrawn after a woman gets a job (e.g. her unemployment benefits) or after the household income surpasses a certain threshold (e.g. means-tested social benefits) [36, 37]. They also cover the taxes or social security contributions which reduce the net additional income she gains from work after increasing her labour supply [38]. The proportion of financial resources taken away from the household relative to the net additional income a woman can earn after she increases her labour supply is commonly referred to as the effective marginal tax rates (MTRs) [39]. By reducing potential household income gains, high MTRs can disincentivize women from increasing their labour market participation. In Europe, MTRs vary by country and within countries in different time periods [40]. They range from around 20% in Greece and Portugal and exceed 50% in Denmark and Germany [40, 41]. Additionally, in countries such as France, Germany, Portugal and Spain, the tax system disproportionately affects women, who face higher MTRs than men in the same earning group [41]. In the light of the above, the final hypothesis this study will test is:

*H5. The labour supply response of women will be weaker in countries with higher marginal tax rates*

## Literature review and research objectives

Previous studies on the response in the labour market supply of women following their partner's becoming unemployed reached conflicting results. Some have concluded that there is no evidence of an increase in women's labour market supply [42], while others have found evidence of such an increase [17–21, 42–46], and others provide mixed results [14, 15].

Many studies have concentrated on the role of individual and household characteristics—couples' age, education, occupation or the presence of children—as explanatory factors. In general, they find that older, better-educated women who have more work experience are more likely to increase their labour supply after their partner becomes unemployed [18, 19, 44, 47, 48]. This is likely because women with better human capital are more attractive in the labour market and thus it may be easier for them to find (additional) employment [49]. Conversely, women with highly educated male partners (i.e. partners with hight income potential) are less likely to increase their labour force participation [18, 19], although this varies based on the country studied [15]. Gender norms can explain these varying results, leading to a more traditional division of labour in couples where men have higher income potentials. Lastly,

although it might be expected that women with non-western migration background are less likely to increase their labour supply, no consistent evidence for such a pattern was found in previous studies [17, 50].

It is less clear how household characteristics, particularly the presence of children, affect women's labour market response after the partner becomes unemployed. In general, an increase in women's labour market participation after their partner becomes unemployed is negatively associated with the number of children [15, 42, 50] and it is particularly low in households with very young children [19, 47, 51]. This is likely because women with young children face time constraints that prevent them from increasing their labour market supply. However, not all studies find evidence for such a negative association [14, 45] and some studies even find a slightly higher increase in the labour supply of wives in households with young children [52].

It seems plausible that these differences stem from the variations in the national childcare policies. For instance, Ghignoni and Verashchagina [51] found that, in Italy, women's response to their partner's unemployment is stronger in regions with higher childcare coverage and in households where grandmothers are present. Conversely, in Austria, a country characterized by a scarce supply of formal childcare, mothers with very young children (under the age of 2) do not increase their labor market supply following the displacement of their husbands [53]. Nevertheless, until now no study has formally investigated the role played by the cross-country variation in childcare policies.

A more limited number of studies have moved beyond investigating the effect of individual and household level characteristics, by focusing on the regional or country-level characteristics which affect women's response. They have largely concentrated on the role of economic conditions, particularly on how women's response differs before, during and after economic crises. During economic crises, households have more difficulty accessing credit [54]. At such times, women may be more likely to increase their labour supply because borrowing money may not be possible to make up for the lost income. On the other hand, during periods of economic downturn, the unfavourable labour market conditions might discourage women from increasing their labour supply [55]. Empirical studies have found support for both expectations. Mattingly and Smith [44] concluded that women in the US were more likely to enter the labour market during the 2008 economic crisis, while the likelihood that they increased their working hours after their partner's unemployment remained relatively constant both before and after the crisis. Conversely, Addabbo et al. [48] concluded that regional unemployment rates were negatively correlated with an increase in the labour market supply of women when their partner became unemployed during the economic crisis. Other comparative studies have examined whether women's response to the unemployment of the male partner varies by country [15], welfare state [16] and unemployment benefit regime [14]. Prieto-Rodriguez and Rodriguez-Gutiérrez [15] investigated women's reactions after their male partner became unemployed in 11 European countries. They found that the reaction depended on the country studied. Women in Italy, Germany, Spain, Portugal and the Netherlands slightly increased their labour market supply, while those in Belgium, Denmark, France, the UK, Greece and Ireland did not. Bredtmann et al. [16] extended this study by comparing women's responses based on categories of the countries' welfare systems. The women's reactions were markedly different based on the broad categories of countries. In Mediterranean countries, women responded to their partner's becoming unemployed by entering the labour market and increasing their working hours, while in Continental Europe the response was limited to an increase in working hours. In Nordic and Central and Eastern Europe the women's response was limited to entering into unemployment from inactivity. Interestingly, in Anglo-Saxon countries women were less likely to become employed when their partner became

unemployed, a fact the authors attributed to the disincentivizing effect of the means-tested benefits in these countries.

McGinnity [14] studied the role of the unemployment benefit regimes, comparing the response of women whose partners had become unemployed in Britain and West Germany, which have considerably different unemployment benefit provisions. Most importantly, Britain means-tests unemployment benefits on family income; Germany does not. McGinnity found that women in Germany were more likely and in Britain less likely to enter employment when their partner lost their job. The authors attributed these differences in behaviours to the differences in unemployment benefit policies. All in all, the notion that welfare policies moderate women's response to their partner's becoming unemployed has been discussed widely in the literature, but no direct test of this hypothesis has been carried out.

Against this background, this paper seeks to expand the understanding of the direct role of specific policies on the labour market response of the female partner after the male becomes unemployed in the 24 EU member states and the UK over an eleven-year period (2009–2019). More specifically, we focus on the role of tax-benefit systems in incentivizing or disincentivizing the women's response and the role that childcare availability plays in enabling women to increase their labour supply in reaction to men's unemployment. As regards the former, we examine the generosity of social transfers and tax policies in replacing the lost income. We also look at the effective marginal tax rates, which may imply a loss of social transfers or an increase in taxes after a woman increases her labour supply. We study the extensive margin of women's response (entry to employment) as well as the intensive margin (transition from part-time to full-time employment) with the latter particularly relevant for current European societies, where women commonly work part-time (though they do so less than men).

## Materials and methods

### Data and sample

The study primarily draws on data from the European Union Statistics on Income and Living Conditions survey (EU-SILC). EU-SILC is a four-year rotational panel, where each country's sample is composed of four sub-samples followed for up to four years (exceptions are France and Norway, which use an eight-year rotational panel). Annually, one sub-sample is dropped and replaced with a new one, thus reducing the problems posed by dropouts [56]. EU-SILC data collection started in 2004 in the EU-15 (except Germany, the Netherlands and the United Kingdom) and in Estonia, Norway and Iceland. The remaining EU-15 countries and NMS-10 (Czechia, Cyprus, Hungary, Lithuania, Latvia, Malta, Poland, Slovenia and Slovakia) joined it in 2005, Bulgaria and Romania and Switzerland in 2007 and Croatia in 2010. The data has been collected annually since then and access to the data is provided by Eurostat upon an acceptance of an official application for the data use [57].

The main advantage the EU-SILC dataset offers is that it provides data which retrospectively measure the monthly activity status of the household members in the year preceding the survey. This enables us to investigate the response in the labour supply of women after their partners became unemployed at a more granular level than would studies which rely on annual data. We restrict the sample to couples surveyed for at least three years, allowing us to analyze employment transitions over a period of two years. We focus on couples in which both are between 25 and 65 years of age, either married or cohabiting and in which the male partner is working in the first month of observation. Out of these couples, we create two subsamples: (a) a sample in which the female partner does not work in the first month of observation (N couple-months: 1.207.260; N couples: 36.601) and (b) a sample in which the female partner works part-time in the first month of observation (N couple-months: 588.492; N couples: 17.037).

Given the need to include only couples with complete monthly work histories, we do not have any observations for Finland, Netherlands and Sweden. As such, our sample includes couples from the other 24 European Union member states and the UK.

In addition to the longitudinalEU-SILC, we estimate childcare availability using the cross-sectional EU-SILC data, that include information on formal childcare use, necessary to test H3. Additionally, we use EUROMOD and UKMNOD, a tax-benefit microsimulation model, and the Hypothetical Household Tool (HHT) it provides. EUROMOD enables cross-country comparability in terms of the effects of taxes and benefits on household income for the countries in the EU. UKMODE offers the same but for the UK. Both allow us to estimate the extent to which social transfers and tax reductions replace the income lost by the male partner with his entry to unemployment, which capture the net replacement rates (NRR), in order to test H4, as well as the marginal tax rates (MRTs), which are needed to test H5. However, the HHT allows us to estimate the NRR and MTRs from 2009 onwards. Lastly, we use Eurostat database and European Values Survey to retrieve country level control variables. Given the data restrictions we focus the analysis on 24 European Union member states and the UK, during the period 2009–2019.

## Prior knowledge of the datasets

Our five hypotheses were derived strictly on the basis of the theory and our research objectives were formulated taking into account past empirical studies. Prior to submitting this manuscript to Plos One, the author responsible for conducting the analyses (Alina Maria Pavelea) had never carried out analyses based on the monthly activity status data offered by EU-SILC. She received official access to the data in September 2022 and started working on it in May 2023 when she used the data solely to compute the sample size, which will be used for the proposed study. One of the co-authors (Anna Matysiak) had previously worked with this dataset, but on different topics and on a limited number of waves. She had not studied women's labour market responses to men's situation in the labour market and had not explored the data on the monthly activity status.

## Research design

### Micro-level work transitions

Table 1 presents a detailed description of the variables considered. As European countries are generally characterized by a high rate of female labour market participation [22], we distinguish between two ways women may increase their labour market supply: entering the labour market or increasing their working hours. We consider two measures of increase in the female labour force participation. First, a transition from being out of work (inactive, unemployed) to being employed, either part-time or full-time (NW→E). Second, an increase in working hours through transitions from part-time to full-time employment (PT → FT). The change in women's labour supply is our main dependent variable and it is coded as 1 if there is an increase and 0 if there is no change. The main explanatory variable is the male partner becoming and remaining unemployed, which is equal to 1 in the months he is considered to be unemployed, and 0 otherwise. As unemployment (state of being unemployed) we consider the situation in which the male partner defines himself as unemployed over the period of three or more consecutive months. This is for two reasons. First, a shorter spell may not require the female partner to increase her labour market supply. Second, short-term unemployment may suggest that it was anticipated by the household, with the male already securing a new job before his employment ended or was terminated.

**Table 1. List and description of variables.**

| Variables | Variable description | Source |
|---|---|---|
| Dependent variables | | |
| NW→E | Binary variable which takes the value of 1 when women increase their labour supply by switching form not working to employment (part-time or full-time) | EU-SILC |
| PT → FT | Binary variable which takes the value of 1 when women increase their labour supply by switching from part-time to full-time work | EU-SILC |
| Explanatory variables | | |
| Unemp | Binary variable which takes the value of 1 when the male partner is in unemployment | EU-SILC |
| Job loss | Binary variable which takes the value of 1 in the months when the male partner has lost his job | EU-SILC |
| Moderator variables | | |
| NRRs | Proportion of household disposable income maintained after the male partner becomes unemployed | HHT EUROMOD |
| MTRs | Proportion of a marginal increase in woman's earnings that is taxed away when women increase their labour supply by switching from a) not working to employment (part-time or full-time) and (b) part-time to full-time work | HHT EUROMOD |
| Childcare 0–3 | Percentage of children aged 0 to 3 enrolled in (part-time < 30 hours/week; full-time = 30 or over hours/week) formal childcare | EU-SILC |
| Childcare 4–6 | Percentage of children aged 4 to 6 enrolled in (part-time < 30 hours/week; full-time = 30 or over hours/week) formal childcare | EU-SILC |
| Childcare 7–12 | Percentage of children aged 7 to 12 enrolled in (part-time < 10 hours/week; full-time = 10 or over hours/week) formal childcare | EU-SILC |
| Household control variables | | |
| Union | Binary variable which takes the value of 1 when the couple is married | EU-SILC |
| Children | Number of children below 18 years of age | EU-SILC |
| Child 0–3 | Binary variable which takes the value of 1 when the couple has a child aged 0 to 3 | EU-SILC |
| Child 4–6 | Binary variable which takes the value of 1 when the couple has a child aged 4 to 6 | EU-SILC |
| Child 7–12 | Binary variable which takes the value of 1 when the couple has a child aged 7 to 12 | EU-SILC |
| Income | Equivalised disposable income (Quintiles) | EU-SILC |
| Individual control variables | | |
| Age | Age in years | EU-SILC |
| Education | Categorical variable: 1) low education (ISCED 1–2); 2) medium education (ISCED 3–4); 3) high education (ISCED 5–6) | EU-SILC |
| Ocuppation | Categorical variable: 1) blue-collar low (ISCO 8–9); 2) blue-collar high (ISCO 6–7); 3) white-collar low (ISCO 4–5); 4) white-collar high (ISCO 1–3) | EU-SILC |
| Country control variables | | |
| Unemployment rate | Quarterly unemployment rate (% of total population 20–64 years) | EUROSTAT |
| Female employment | Quarterly female employment rate(% of total population 20–64 years) | EUROSTAT |
| Women gender role attitudes | Percentage of women by birth cohort who agree with the statement: "men should take the same responsibility as women for children and home" | EVS |
| Men gender role attitudes | Percentage of men by birth cohort who agree with the statement: "men should take the same responsibility as women for children and home" | EVS |

The NRRs for the continuously employed male partners this takes the value of 100.

In order to better explain our research design Fig 1 depicts examples of possible employment transitions of the couples. Our focus is on couples in which the man is employed and the female partner is not working or employed part-time in the first month they are surveyed (t1). During the time of observation the woman can either experience no change in her labour supply, can increase it or her labour supply can decline (if a woman changes from part-time into non-employment). We observe the couple until the last month they are followed in the survey (tn) or until the man becomes inactive, a woman's labour supply declines or the union dissolves, whichever comes first.

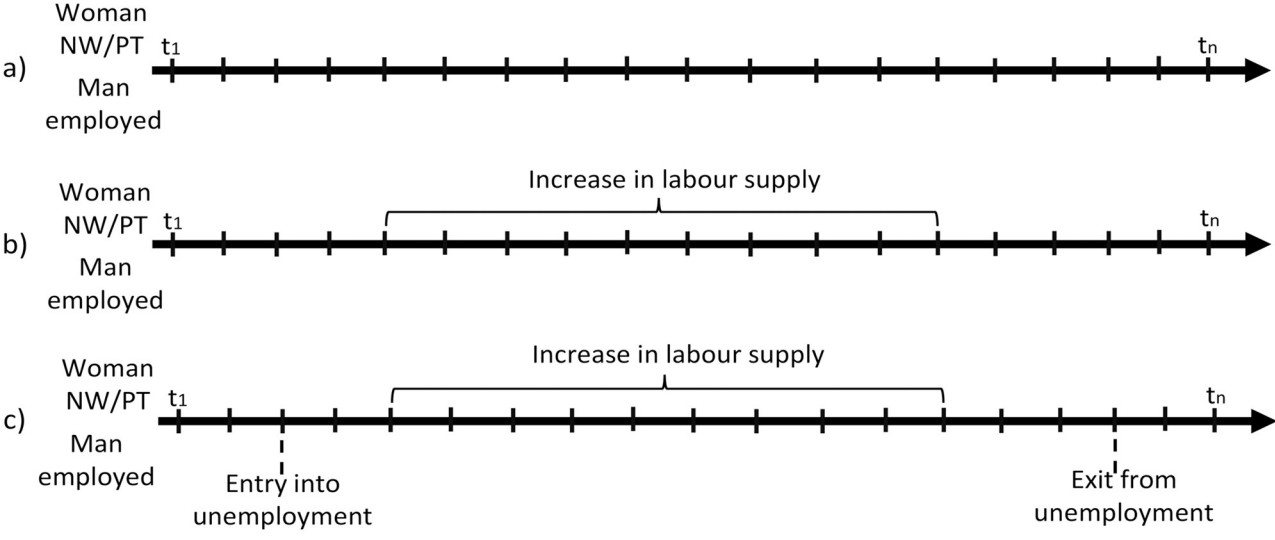

**Fig 1. An example of the possible employment transitions of the couples.**

## Country-level indicators

First, in measuring childcare availability we consider three age groups, based on the type of formal childcare they require (0 to 3 years of age, 4 to 6 years of age, and 7 to 12 years of age) and include three types of formal childcare (childcare at day-care center, education at pre-school, and childcare at center-based services outside school hours). Given its nature, we do not include education at compulsory school, as there is little variation in enrollment across the studied countries. By making use of the EU-SILC cross-sectional data we compute multiple alternative measures of childcare availability: the percentage of children enrolled in formal childcare out of the corresponding age population; the percentage of children enrolled part-time in formal childcare; the percentage of children enrolled full-time in formal childcare. We use childcare enrollment because it is a widely used measure of availability [58, 59].

Next, using the Hypothetical Household Tool (HHT) in EUROMOD and the UKMOD, we measure the extent to which social transfers and tax reductions replace the income lost by the male partner upon becoming unemployed (NRRs) and the proportion of the household income that is taxed away when the woman enters employment relative to the net additional income a woman earns after she increases her labour supply (MTRs). We employ the HHT as the EU-SILC longitudinal data we are using is not compatible with EUROMOD and UKMOD. Although EUROMOD/UKMOD can be used with EU-SILC microdata, it relies on the cross-sectional EU-SILC data, which cannot be linked with the longitudinal EU-SILC data, according to the Eurostat's principles of micro data use.

As mentioned, EUROMOD and UKMOD are tax-benefit simulation tools which store information on the tax and benefit policies of EU countries and UK—respectively—in specific time periods. As such, it can be used to estimate the consequences tax and benefit policies have on net household incomes. For such an estimation to be possible, detailed data on household composition, earnings of household members and their demographic characteristics are required. We use the HHT to generate hypothetical household data which will later be input-ted into EUROMOD and UKMOD in order to estimate household income after taxes and transfers. Generating multiple types of identical households, which only differ in the

employment situations of the partners, will allow us to estimate the NRRs and MTRs. The NRRs (for specific countries and periods) can be computed by relating the net household income of two identical hypothetical households that only differ in the employment situation of the male partner (employed vs. unemployed), as follows:

$$\text{N}RR = \frac{Y_{HH}^{U}}{Y_{HH}^{E}} * 100, \tag{1}$$

where $Y_{HH}$ represents the household disposable income when the male partner is in unemployment, while $Y_{HH}^{E}$ represents the household disposable income when the male partner is employed. Likewise, the MTRs are assessed as a complement of the ratio of (a) the difference in net household income of two hypothetical households which differ in the employment situation of the female partner but are identical otherwise and (b) a difference in woman's gross earnings after she increases her labour supply:

$$\text{M}TR = \left( 1 - \frac{Y_{HH}^{1} - Y_{HH}^{0}}{E_{i}^{1} - E_{i}^{0}} \right) 100 \tag{2}$$

where $Y_{HH}$ represents the household disposable income before (0) and after (1) woman's labour supply increase, while $E_{i}$ represents the gross earnings of the woman before (0) and after (1) the labour supply increase. In order to estimate the NRRs and MTRs for couples that resemble as much as possible the couples in our EU-SILC dataset, we generate a large number of hypothetical households, based on various characteristics. In generating hypothetical household data, the HHT requires users to specify the household composition by defining the relationship between the household members (e.g. single/cohabiting, couple with children, couple without children) and the demographic characteristics of the household members. For example, the user can specify, for all household members, the age, economic status (employed, unemployed, inactive), number of hours worked and income from employment (as % of EU-SILC average). For these variables users can also specify the values between which the variables range and the range itself. We take the following variables into account when generating hypothetical data for computing the NRR and MTRs: the marital status of the couple (married or cohabiting), the employment status of the men and women and the number and age of the children. As the number of households generated grows exponentially with the number of categories assumed by each variable we aimed at reducing the number of the categories to a minimum.

S1 Table. Hypothetical Household characteristics for calculating income replacement rates offers an overview of the hypothetical households generated to estimate the NRRs. In this way we estimate the replacement rates after the male becomes unemployed, which differ in the employment situation of the female partner, who can be inactive or unemployed (HH1), and employed part-time (HH2). We also consider the union type (cohabiting, married) andthe number of kids and their age, as household composition influences the net household income [35]. We limit ourselves to generating households with up to 3 children, as most couples have a maximum of 3 children [60], and we range the age of the children from 0 to 18 years in increments of 3 years. The NRRs corresponding to households with 3 children will also be assigned to households with more than 3 children.

S2 Table. Hypothetical Household characteristics for calculating MTRs offers an overview of the hypothetical households generated to estimate MRTs. Here we generate identical households that differ in the employment situation of the female partner in T0 and T1. This allows us to estimate the MTRs associated with the two employment transitions under consideration: NW→E and PT → FT. All couples in EU-SILC are assigned country-period specific MTRs

irrespectively whether they increase their labour market supply or not. We assign them two sets of MTRs, one corresponding to transitions from a) not working to part-time employment and b) part-time employment to full-time employment, and in estimating our models we consider both. We generate hypothetical households where the husband is unemployed (HH1) or employed (HH2). The latter values are assigned to the EU-SILC couples where the husband is continuously employed, as their wives might also transition into employment.

When using the HHT, a series of assumptions have to be made at the individual and household level. When generating hypothetical data to estimate NRRs we assume that the men's employment income is 33% of the country-year average in part-time employment and 67% of the country-year average in full-time employment. In a similar fashion, to estimate MTRs we set women's wage to 33% of the country-year average for part-time employment and 67% of the country-year average for full-time employment. Although the HHT allows users to also specify expenditures the hypothetical households might have (e.g. rent), we do not use this option. This ensures that the differences in the net household income of hypothetical households that differ only in the partners' employment situation strictly reflect the role of tax and benefit policies. Lastly, EUROMOD and UKMOD assume 100% benefit take-up. It is highly likely that couples rather make use of the social transfers they are eligible to, although this assumption may lead to a small overestimation of the replacement rates and MTRs [37].

## Control variables

Apart from the main variables of interest, we control for a large number of factors that can play a role in the labour market response of women to their partner's unemployment. At the individual level, we control for the age and education of the partners. Regarding occupation, we control for the occupation of the male partner in all models, whilewe also control for women's occupation when investigating women's transition from part-time to full-time employment. At the household level, we account for the type of union, income, number of children and age of the children. At the country level, we consider the quarterly unemployment rate and quarterly employment rate of females. As mentioned above, high unemployment rates might discourage women from increasing their labour supply [50]. Additionally, in countries with high levels of female employment there is little space for an increase in women's labour supply in response to the unemployment of male partners. In fact, previous studies found women's response at extensive margin to be weak in such countries [43, 44].

Lastly, we also consider the predominant gender-role attitudes in the country, as studies show that egalitarian gender-role attitudes at the country level are positively associated with female employment [61]. However, time series data on gender-role attitudes comparable for countries covered by our study are not available. In order to solve this problem we follow Briselli and Gonzalez [62] who propose to construct country-cohort gender-specific measures of gender role attitudes using data from the European Values Survey 2008 (EVS). Using EVS we compute the country-cohort and gender-specific proportion of respondents who agreed with the statement "men should take the same responsibility as women for children and home" shortly before our observation period (which starts in 2009). In this way, we capture the variation in social acceptance of men's involvement in childcare and housework across cohorts and countries. This approach assumes variation in gender role attitudes across cohorts, but not within cohorts over time. Although multiple studies found evidence of differences in gender role attitudes across cohorts, there is less evidence that these gender role attitudes are stable within cohorts [63]

## Model specification

The aim of the study is to estimate the probability of specific employment transitions by females after their partners have become unemployed and the moderating effect of policy variables. We account for the hierarchical structure of our data, with individuals observed in multiple time periods, nested within countries, leading us to employ a multilevel model. More specifically, we estimate mixed effects Linear Probability Models, with random slope at the country level and robust standard errors. The main advantages of Linear Probability Models (LPM) is that they are intuitively meaningful, as their results can be interpreted as differences in probabilities [64]. Although the use of LPM has been criticized because a predicted probability may fall outside the range 0–1, simulations show that the predicted probabilities from linear and logistic regression analyses are nearly identical [64], the only difference residing in the ease of interpreting the results of the former. To tests the first hypothesis we estimate the following baseline model:

$$y_{ictm} = \beta_0 + \beta_1 Unemp_{ictm} + \delta' X_{ict} + \phi' P_{ict} + \gamma' Z_{ict} + U_c + \varepsilon_{ictm} \tag{3}$$

where $y_{ictm}$ is the dummy variable on women's labour supply increase, which is equal to 1 if the woman i in country c in year t, month m increased her labour market supply by transitioning from not working to employment (NW→E), in the first specification, or from part-time to full-time work (PT → FT), in the second specification. $Unemp_{ictm}$ is the dummy variable on the male partner's employment status, which equals 1 during the months when the male partner is unemployed. The vector of covariates $\delta' X_{ict}$ includes the partners' and household characteristics, which vary by year. The vector $\phi' P_{ict}$ includes the policy variables of interest. As including all policy variables in the model can run the risk of multicollinearity, we will include the policy variables either separately or jointly depending on the VIF. $\gamma' Z_{ict}$ represents the county-level control variables, and $U_c$ is the country-random effect. To test our other hypotheses we include interaction terms between $Unemp_{ictm}$ and the moderator variables of interest, which take the form:

$$y_{ictm} = \beta_0 + \beta_1 Unemp_{ictm}*Child0-3_{ict} + \beta_2 Unemp_{ictm}*Child4-6_{ict}+ \tag{4}$$

$$\beta_3 Unemp_{ictm}*Child7-12_{ict} + \delta' X_{ict} + \phi' P_{ict} + \gamma' Z_{ict} + U_c + \varepsilon_{ictm} \tag{5}$$

$$y_{ictm} = \beta_0 + \beta_1 Unemp_{ictm}*NRRs_{ictm} + \delta' X_{ict} + \phi' P_{ict} + \gamma' Z_{ict} + U_c + \varepsilon_{ictm} \tag{6}$$

$$y_{ictm} = \beta_0 + \beta_1 Unemp_{ictm}*MTRs_{ictm} + \delta' X_{ict} + \phi' P_{ict} + \gamma' Z_{ict} + U_c + \varepsilon_{ictm} \tag{7}$$

To test the second hypothesis we interact $Unemp_{ictm}$ with the dummy variables $Child0-3_{ict}$ $Child4-6_{ict}$ and $Child7-12_{ict}$, which take the value of 1 when the couple has a child in these age categories. We test the third hypothesis by expanding this model and include the interaction with childcare availability for each age group. Lastly, we test hypotheses 4 and 5 by interacting $Unemp_{ictm}$ with $NRRs_{ictm}$ and $MTRs_{ictm}$, respectively. We use a significance level of 5% as inference criteria for our hypotheses.

## Robustness and sensitivity analyses

Because households may anticipate unemployment, which can affect their response [65], we have also considered narrowing the analysis to involuntary job losses, as they are less likely to be foreseen. Given the lower risk that involuntary job losses (i.e. when the employee is made redundant by the employer)are anticipated [53], in this case we consider all unemployment

spells regardless of duration. In identifying a job loss, we combine the monthly activity status variables with data from two other questions: if the respondent had changed their job since the last interview (in the previous 12 months) and the reason for that change that they were "obliged to stop by the employer". Thus, when the respondent experienced an unemployment spell over the previous 12 months and reported having changed his job because he was "obliged to stop working by the employer", we assume that the unemployment spell was as a result of a job loss. Here we would stress, however, that focusing on job loss may result in the problems that accompany small sample sizes. Not having been able to work with the dataset prior to undertaking the study, we cannot be sure if we will achieve this aim, and thus we treat this as a supplementary analysis. Additionally, research shows that an increase in the labour supply of women is most likely to occur when the male partner has been unemployed between three and six months, but not significant when the duration of the man's unemployment is longer [20]. Arguably, this might be because long unemployment spells cause households to adapt to the loss of income in other ways, such as by reducing their consumption [66]. As such, we will test the sensitivity of the results by considering only unemployment spells that last between three and six months, and unemployment spells that last for more than six months—separately. We test the sensitivity of our results by considering different specifications for the policy variables of interest. More specifically, we estimate the NRRs and MTRs by using different assumptions concerning the partners' wage. In this sense, we set the wage of part-time and full-time employment to: a) 25% and respectively 50% of EU-SILC average wage and b) 50% and respectively 100% of EU-SILC average wage.

## Supporting information

**S1 Table. Hypothetical household characteristics for calculating income replacement rates.**
(DOCX)

**S2 Table. Hypothetical household characteristics for calculating MTRs.**
(DOCX)

## Author Contributions

**Conceptualization:** Anna Matysiak, Anna Kurowska.

**Data curation:** Alina Maria Pavelea.

**Funding acquisition:** Anna Matysiak.

**Methodology:** Anna Matysiak, Anna Kurowska, Alina Maria Pavelea.

**Supervision:** Anna Matysiak, Anna Kurowska.

**Writing – original draft:** Anna Matysiak, Anna Kurowska, Alina Maria Pavelea.

**Writing – review & editing:** Anna Matysiak, Anna Kurowska, Alina Maria Pavelea.

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
