## [Decision Letter · Decision Letter 0]

11 Jul 2023

PONE-D-23-15097His unemployment, her response, and the moderating role of welfare policies in European countriesPLOS ONE

Dear Dr. Matysiak,

Thank you for submitting your manuscript to PLOS ONE. After careful consideration, we feel that it has merit but does not fully meet PLOS ONE’s publication criteria as it currently stands. Therefore, we invite you to submit a revised version of the manuscript that addresses the points raised during the review process.

As you will see the paper prompted a lot of interest and we have 7 referee reports. Four of the referees have asked for Major Revisions and three for Minor Revisions. Most of the comments ask for clarifications and some comments ask for additional work. Please address the clarification issues and the issues requiring additional analysis, if feasible, with the dataset you have already been using. If some of the requested additions are not feasible, please explain why.

We look forward to receiving your revised manuscript.

Kind regards,

Daphne Nicolitsas

Academic Editor

PLOS ONE

Journal Requirements:

2. In your cover letter, please confirm that the research you have described in your manuscript, including participant recruitment, data collection, modification, or processing, has not started and will not start until after your paper has been accepted to the journal (assuming data need to be collected or participants recruited specifically for your study). In order to proceed with your submission, you must provide confirmation.

"This project has received funding from the European Union’s Horizon Europe research executive agency (REA) under Grant Agreement No.101060410. Views and opinions expressed are however those of the author(s) onlyand do not necessarily reflect those of the European Union or [name of the granting authority]. Neitherthe European Union nor the granting authority can be held responsible for them"

7. Please ensure that you include a title page within your main document. You should list all authors and all affiliations as per our author instructions and clearly indicate the corresponding author.

Additional note from the journal: Please note that Reviewer #7 has commented on the lack of results from this study - as these concerns are not applicable in the case of your particular study type, you are not obligated to directly address this point in your revisions. 

Reviewers' comments:

Reviewer's Responses to Questions

**Comments to the Author**

1. Does the manuscript provide a valid rationale for the proposed study, with clearly identified and justified research questions?

Reviewer #1: Yes

Reviewer #2: Yes

Reviewer #3: Yes

Reviewer #4: Yes

Reviewer #5: Yes

Reviewer #6: Yes

Reviewer #7: Yes

2. Is the protocol technically sound and planned in a manner that will lead to a meaningful outcome and allow testing the stated hypotheses?

Reviewer #1: Yes

Reviewer #2: Partly

Reviewer #3: No

Reviewer #4: Yes

Reviewer #5: Yes

Reviewer #6: Yes

Reviewer #7: Yes

3. Is the methodology feasible and described in sufficient detail to allow the work to be replicable?

Reviewer #1: Yes

Reviewer #2: Yes

Reviewer #3: No

Reviewer #4: Yes

Reviewer #5: Yes

Reviewer #6: Yes

Reviewer #7: Yes

4. Have the authors described where all data underlying the findings will be made available when the study is complete?

Reviewer #1: No

Reviewer #2: Yes

Reviewer #3: Yes

Reviewer #4: Yes

Reviewer #5: Yes

Reviewer #6: No

Reviewer #7: Yes

5. Is the manuscript presented in an intelligible fashion and written in standard English?

Reviewer #1: Yes

Reviewer #2: Yes

Reviewer #3: Yes

Reviewer #4: Yes

Reviewer #5: Yes

Reviewer #6: Yes

Reviewer #7: Yes

6. Review Comments to the Author

You may also provide optional suggestions and comments to authors that they might find helpful in planning their study.

Reviewer #1: Overall, this is an excellent report. It has a nice, clear title, a clear literature review, and provides detail on the methodological approach.

I have some comments which I hope the authors find useful.

The focus on childcare availability seems to underplay the importance of cost (and possibly quality). It may not pay for the woman to take a job, especially a low-paid one, if most of her wages will be wiped out by childcare costs. Perhaps in these couples the man HAS to take on the care work, otherwise the family will not survive, and hence the availability of childcare services is largely immaterial to the added worker effect. H5 discusses the financial ‘costs’ of working, but does not include childcare among these.

This also links to differences by families’ and the mother’s socioeconomic status. It would be interesting to see how the added worker effect varies by the man’s/woman’s educational level – rather than simply controlling for this, it might be interesting to compare for lower-educated versus highly-educated.

How does ECEC interact with parental leave? Some countries with lower enrolment in childcare for under 3s – one of the country-level indicators – also have paid leave available, although of course this varies by payment level and duration. Parental leave seems an important policy that should be accounted for in the study.

I also wondered about school schedules and the availability of wrap-around care for school-aged children, which vary greatly across countries. In addition, what about elder care services?

I also wonder about the role of culture. As research has shown (e.g., Budig et al., 2012), culture is an important moderator of the impacts of policies on women’s employment.

Relatedly, another important control variable omitted is an individual’s gender-role attitudes. We might expect more traditional attitudes to hamper the added worker effect.

The report mentions on p.15: “when investigating transitions from part-time to full-time employment, we stop observing the couple if the woman becomes inactive or unemployed.” Can you give more detail as to why?

The section on ‘Robustness and Sensitivity Analyses’ isn’t entirely convincing. The testing of ‘involuntary’ job loss is somewhat confusing, since you are focusing on unemployed men only. Unemployment is surely rarely ‘voluntary’ and chosen, if ever. The report mentions ‘involuntary’ unemployment will be operationalised by if the respondent had changed their job because they were “obliged to stop by the employer”. It might be useful to state what other reasons can be given and how these might indicate ‘voluntary’ or ‘planned’ unemployment, as the report implies.

I was left wondering whether you are actually measuring male unemployment or male joblessness defined more broadly as any transition out of employment.

Additionally, assuming that restricting the analysis to people aged 25-45 will give insight into the role of liquidity constraints seems tenuous. There is likely significant heterogeneity in such constraints within this age group.

Can you do another sensitivity analysis looking at when the man is unemployed >6months?

I think it would be good to include a research question.

There are some language errors that will need correction before publication. E.g., in the abstract, “the” should be removed from “Recent changes in the labour markets”.

Reviewer #2: The scope of the study is to investigate the potential increase of women labour supply once man become unemployment. The value added of the study is to also consider the mediating role of the policy environment. I think the proposed methodology can partially answer to the research question but I have a few doubts.

1. The dependent variables proposed are of two kinds, one, women move from unemployment to employment, two, women move from part-time to full time. I wonder whether the data allow to also have a more sophisticated measure considering also eventual additional hours (both in terms of women increase, but also of men lost, rather than simply the movement to unemployment). This would allow to calculate a sort of elasticity in terms of hours of work reduction from men and increase for women within a couple (and of course the mediating role of politics).

2. Is EU-Silc the best dataset for this? What about the labour force survey for looking in more detail at the labour market characteristics of men and women?

Reviewer #3: This registered report is written in clear and concise language, and centres on an interesting research question. Specifically, the report highlights how the researchers aim to analyse the intervening impact of tax benefit policies on the one hand and child care policies on the other hand, on a spouse’s response to her husbands unemployment. The researchers succeed in highlighting why this question is relevant, and have a sound theoretical basis for their hypotheses.

However, the report describes a number of odd methodological choices, that lead me to doubt whether the proposed research will succeed in reaching its professed aims. In addition, a number of additional operationalization and methodological choices are not outlined in sufficient detail to assess at this stage whether the research will be feasible and replicable.

- The authors cite the total sample size, in terms of couple-months and couples. However, in order to assess the feasibility of the study, more information is needed: how large are the sample sizes by country, especially when also considering the numbers of “relevant” couples, i.e. couples where the male makes a transition from employment to unemployment within the period studied, per observation period.

- Including the transition from studying to employment (for the spouse) seems not in line with the proposed research question.

- It is not entirely clear what the observation period is, for how long and which couple employment transitions are included. It may help to extend figure 1 with a number of additional examples, in terms of promoting replicability.

- My main issue with the report is the proposed approach to take account of tax benefit policies. The authors propose to derive hypothetical NRRs and MTRs for a large share of hypothetical households using the hypothetical household tool from EUROMOD, in order to assess how the incentives of the tax benefit policies impact on the spouses response to her husband’s unemployment spell. However, the main advantage of using EUROMOD would precisely be to be able to calculate at the individual level the actual NRRs and MTRs faced by each individual, both in the affected households as for the unaffected households. Given that they clearly are aware of the possibilities of EUROMOD, I see no reason why the authors would opt for the less precise operationalization of the impact of the tax benefit system in their approach. I can imagine that using actual incentives might make it less self-evident to use a country-level assessment of the tax benefit policies, but in that case the methodology should be changed, rather than choose a second-best option to fit the model specification.

- The choice to assess availability to child care by looking at coverage rates is defensible. However, as the authors themselves highlight the main reason for this choice is pragmatic, an obvious robustness check would be to include the SILC information on current use of formal and informal child care, especially in the parttime case. Getting access on a relatively short term notice (only 3 months after all) is definitely at odds with the reality in a number of European countries. One could assume that it is easier to extend child care hours, than to claim a new spot altogether on short notice.

Finally, I have a number of smaller remarks regarding the literature review.

- The references cited on the added work effect are relatively dated, or, alternatively, focus on Turkey or Latin-American countries.

- The literature review only hints at two very likely determinants of differences in the added worker effect at the household level: cultural preferences, that may differ between countries, but certainly also within countries, by social class or by minority group. The literature review is not explicit about this, and the research also fails to control for these determinants in the proposed methodology. Earnings potential and education is included in the model, but its likely impact is only summarily addressed in the literature review.

- The increase in men’s employment instability is given as the main rationale for embarking upon the proposed research. However, the reasons for this increased instability leads one to wonder why this would only be a “male” phenomenon. Why would the same drivers have a different impact on female employment? Also, the instability driven by a higher share of fixed employment contracts would not necessarily translate in more mid-term unemployment spells, if they are conducive to a well-functioning labour market. Insiders and outsiders are likely affected differently.

Reviewer #4: The report protocol describes a study which aims to investigate women’s labour market choices when their partners have become unemployed, and whether welfare state policies modify their responses, focusing on childcare and tax/benefit policies. The topic is highly relevant, and the study has a potential to provide valuable new information on partners’ interlinked labour market choices in couples.

The aim and design of the study are adequately described. The protocol provides relevant literature and discusses research gaps, which their study will address. Hypotheses (H1-H5) are clear, and can be investigated with the chosen methodology and data.

Data (EU-SILC) are sufficient for conducting the planned study. The study aims to use Eurostat information on childcare coverage and EUROMOD/UKMOD simulation tools to provide information on the welfare policies.

I have some suggestions for the authors to consider:

Country-level indicators

The study aims to examine the role of taxation/benefits and childcare policies in potentially modifying women’s responses to their partner’s unemployment. However, I wonder if it is eventually possible to distinguish the role of NRR, MTR, or childcare availability if they go hand in hand, e.g. a country with high childcare availability may also be a country with high MTR (for example, the Nordics). I take that this remains to be tested empirically, but may be worth considering already before analyses, and to avoid overstating the conclusions. There could also be other (policy) factors than those examined in the analyses which could explain country-level variation in women’s responses. For example, if (women’s) part-time work is very common in a country, transition from inactivity to (part-time) employment may be easier even with small children.

Analyses of the transitions

It was unclear how the male partner’s return to employment (after a period of unemployment) is treated in the analyses.

Also, how is union dissolution taken into account – are couples censored at the time of dissolution? (and at what precision – monthly, annually – is the time of union dissolution measured?)

Calculations of NRR/MTR

The authors treat partners in cohabiting couples as singles when calculating NRR and MTR. I take that single here means single(parent)+children, when cohabiting couples have children, and not a single-person household (?).

If I understand correctly, treating cohabiting couples differently from married couples is based on how they are treated in national taxation and benefit policies. I wonder, however, if in many countries cohabiting couples resemble married couples (in terms of taxation and many benefits) when they have common children? E.g. in the case of cohabiting couple with children living in the household, would the ‘match’ in EUROMOD/UKMOD be a married couple with children, rather than a male/female single adult (with children)?

I assume that the NRR and MTR strongly depend on personal/household income. For example, MTR varies by income level, and is generally higher at lower income levels (at least in countries which provide various means-tested benefits for families). Using the average EU-SILC wage to calculate disposable income in reference households ignores this aspect, if I understood the procedure correctly (that the authors are assigning an average country-level wage to households with full-time employed male/female partner). If my interpretation is correct, would it be meaningful to calculate MTR in the lowest /two lowest income quintiles (income quintiles calculated from the EU-SILC data)?

Childcare statistics

While statistics on childcare coverage (participation rates) are used to measure childcare availability, they are not optimal as in some countries the participation rates of children aged 0-2-years are low not due to low availability of ECEC services for toddlers, but (partly) due to that parental leaves or extended parental leaves allow parent(s) to stay at home longer (for example, Estonia, Malta, Hungary, Finland, see LeaveNetwork Country reports and Cross country reports on ECEC, https://www.leavenetwork.org/introducing-the-network/). I do not suggest to use other sources, however, as the Eurostat participation rate statistics are superior in the country comparison.

Reviewer #5: Dear authors. Thank you for submitting the protocol to the editorial office. I have only two comments about the manuscript.

Please specify why you chose the Linear Probability Model (LPM) for modeling and not the Logistic Model or Probit Model more commonly used in similar situations.

What assumptions does LPM have and does your data meet them?

Reviewer #6: Overall, I find the research problem topical and important from both a theoretical and practical point of view. The moderating role of childcare and tax and benefit policies on the labour supply response of women to the unemployment of their partners has not been thoroughly nor empirically studied and research in this area could lead to relevant findings and proposals for policies.

I find the hypotheses to be in line with the general research objective and grounded on sound theoretical foundations. The literature review briefly describes the most recent and relevant findings in the literature and is well-connected to the hypotheses.

Data and variables are described in small-enough detail for the reader to understand the methodological design of the study. However, the equations for the models that will be used to test the different hypotheses should also be provided and explained. Some details would also be welcome on the reasons for choosing each control variable that is added to the model, supported by other studies using the same or similar variables. The authors should provide some information on where the data will be made available when the study will be completed.

Some propositions for robustness checks have been formulated. I suggest the authors to also look at the opportunities offered by the European Commission’s Tax and Benefits Indicators database for additional variables for robustness checks.

Last but not least, although I appreciate the overall quality of the report as very high, further attention to some minor details is required. For example, I would see the research objectives presented in the introduction, before formulating the research hypotheses. There are some inconsistencies in the text regarding the number of countries in the sample (24 EU Member States on p. 8 – nothing about the UK, and 25 European countries on p. 13). Moreover, the period 2009-2019 covers 11 years and not 10, as stated on p. 13.

I congratulate the authors for their research plans and wish them good luck and good results!

Reviewer #7: - In the introduction section you should present the methods used for analysis, the results obtained and also the usefulness of your results

- the results and discussion section is missing. In the section Model specification, you talk about using regression to test the hypothesis, but you do not present the effective results. You should present some tables with the results. Have you effectively analysed the data? You used only regression? This part is not clear.

- The conclusion section is missing. This section should contain the main results of your paper and the value added generated by your study. Also, you should describe which are the limitations of your paper.

7. PLOS authors have the option to publish the peer review history of their article (what does this mean?). If published, this will include your full peer review and any attached files.

Reviewer #1: No

Reviewer #2: No

Reviewer #3: No

Reviewer #4: **Yes: **Anneli Miettinen

Reviewer #5: No

Reviewer #6: **Yes: **Irina Bilan

Reviewer #7: No

---

## [Author Response · Author response to Decision Letter 0]

15 Sep 2023

Reviewer #1: 

Overall, this is an excellent report. It has a nice, clear title, a clear literature review, and provides detail on the methodological approach.

Response: Thank you!

I have some comments which I hope the authors find useful. 

The focus on childcare availability seems to underplay the importance of cost (and possibly quality). It may not pay for the woman to take a job, especially a low-paid one, if most of her wages will be wiped out by childcare costs. Perhaps in these couples the man HAS to take on the care work, otherwise the family will not survive, and hence the availability of childcare services is largely immaterial to the added worker effect. H5 discusses the financial ‘costs’ of working, but does not include childcare among these.

Response: 

Thank you for this comment. Indeed, we agree that the cost of childcare is of relevance. There are two reasons we do not specifically include it. First, there is the problem of availability of longitudinal data on childcare costs for the countries and age categories of our interests. For example, although OECD provides data on childcare cost, it does not cover the entire period of observation and it only concerns one age category of children (below 3). Second, we think that enrollment in childcare partially captures childcare costs. In countries where childcare costs are high, the coverage is lower precisely because some people cannot afford childcare. 

This also links to differences by families’ and the mother’s socioeconomic status. It would be interesting to see how the added worker effect varies by the man’s/woman’s educational level – rather than simply controlling for this, it might be interesting to compare for lower-educated versus highly-educated.

Response: 

Indeed , the added worker effect can vary based on the education level of the man/woman. In fact research on this topic has been already conducted (we mention some of these studies in our paper in section Literature Review) which is why we initially decided not to include the AWE by mothers’ / partners’ education into our study. We agree that it would be very interesting and novel to study the moderating effect of the policies on AWE by women’s / partner’s education. That would, however, substantially expand the paper, making it too long and too complex. We think that such a study could be conducted as a next step and presented in a separate paper and we will be happy to do it. 

How does ECEC interact with parental leave? Some countries with lower enrolment in childcare for under 3s – one of the country-level indicators – also have paid leave available, although of course this varies by payment level and duration. Parental leave seems an important policy that should be accounted for in the study. 

Response: 

As you mentioned, countries with lower enrolment in childcare for under three years of age may offer (generous) paid parental leaves instead. In these countries households may thus not rely on childcare, but rather on paid parental leaves. This, however, should not influence the relationships we are studying. Our focus is primarily on the labour supply response of those women, who are not employed. And their reaction is to the unemployment of their male partners.. In most European countries neither the non-employed mothers nor the unemployed fathers are eligible for paid parental leaves (parental benefits) . This is one of the reasons that reduce the relevance of the availability of paid parental leave for this particular study. The only group that we study, that could be entitled to paid parental leave would be mothers working part-time. However, it is not very likely that these women would opt for going for a parental leave when their partner becomes unemployed. In the majority of European countries, paid parental leave arrangements are provided during the first 5-12 months after the child is born. Therefore this should not be relevant for the vast majority of our sample. Longer parental or childcare leaves available for parents are mostly unpaid and therefore these should not matter for the maternal incentives to increase their labour supply. 

I also wondered about school schedules and the availability of wrap-around care for school-aged children, which vary greatly across countries. In addition, what about elder care services?

Response: 

Indeed the availability of formal childcare might be important for mothers with older children as well, such as school-aged children. In line with this suggestion, we have adjusted hypotheses 2 and 3, so as to include mothers, in general. The data source we planned to use before (Eurostat) to test the hypotheses 2 and 3 offers data on formal childcare enrollment only for children aged less than 3 or between 3 and compulsory school age (data code: TPS00185) and thus misses school-aged children. For this reason we decided to change the data source and use cross-sectional EU-SILC data to compute measures of enrollment in formal childcare at the national level for three age groups, based on the type of childcare they require: 0 to 3 years of age, 4 to 6 years of age, and 7 to 12 years of age (see manuscript page 9). In this way we are also able to distinguish between the general percentage of children enrolled in formal childcare and the percentage of children enrolled full-time in childcare.

Regarding elder care services, the EU-SILC data does not allow us to identify whether the respondent has a care responsibility over an elderly person. As such, in this instance we cannot test the moderating role of the availability of elder care services, which should matter only to people with such responsibilities. 

I also wonder about the role of culture. As research has shown (e.g., Budig et al., 2012), culture is an important moderator of the impacts of policies on women’s employment.

Relatedly, another important control variable omitted is an individual’s gender-role attitudes. We might expect more traditional attitudes to hamper the added worker effect.

Response: 

We fully agree that both individual gender-role attitudes and cultural norms can influence the added-worker effect. Unfortunately, we are not able to control for the individual gender-role attitudes, as the EU-SILC data does not include any questions on this topic. Moreover, we were not able to find any time series on cultural gender norms for the countries and time period we study. After your comment, we thought about this problem a bit more and came up with the idea to compute a country-cohort and gender-specific measures of gender role attitudes following Briselli and Gonzalez (2023). They make use of the European Values Survey (EVS) 2008 and the agreement of respondents for two statements: “Sharing house chores is very important for a successful marriage” and “men should take the same responsibility as women for children and home” to measure men’s and women’s views on family roles in the household., We decided to follow their approach and we plan to compute the share of men and women in different birth cohorts who agreed with the statement: “men should take the same responsibility as women for children and home”, as we are interested in the attitudes concerning both housework and childcare (see manuscript page 12). This approach will allow us to control for the variation across cohorts and countries in gender role attitudes before our observation period, i.e. in 2008. 

The report mentions on p.15: “when investigating transitions from part-time to full-time employment, we stop observing the couple if the woman becomes inactive or unemployed.” Can you give more detail as to why? 

Response: 

We stop following couples if the woman becomes inactive or unemployed as the transition we are interested in (from part-time to full-time) is not possible in their case. 

The section on ‘Robustness and Sensitivity Analyses’ isn’t entirely convincing. The testing of ‘involuntary’ job loss is somewhat confusing, since you are focusing on unemployed men only. Unemployment is surely rarely ‘voluntary’ and chosen, if ever. The report mentions ‘involuntary’ unemployment will be operationalised by if the respondent had changed their job because they were “obliged to stop by the employer”. It might be useful to state what other reasons can be given and how these might indicate ‘voluntary’ or ‘planned’ unemployment, as the report implies.

I was left wondering whether you are actually measuring male unemployment or male joblessness defined more broadly as any transition out of employment.

Response: 

The use of the word involuntary might have been confusing, we thus removed it (see manuscript page 13-14). We are not claiming that unemployment is voluntary, but rather that in some cases it can be anticipated by the household (see Stephens, 2002) for example, because the work is seasonal or because a temporary contract ends. When unemployment is anticipated, a woman may increase her labour supply already before the man enters unemployment. A job loss (forced redundancy by employer) , on the other hand, is less likely to be anticipated and if a woman decides to increase her labour supply she will rather do so after the job loss has happened. All in all, our study may underestimate the AWE and that is why we decided to conduct this robustness check to verify whether this is the case and what is the magnitude of this underestimation. Unfortunately, we can do it only if the number of observations of identifiable job losses remains sufficient for estimating the models. 

With regards to the second comment, we identify unemployment based on the self-defined economic status, as this is the only approach possible based on the EU-SILC retrospective monthly data. As such, this variable might capture both unemployment and joblessness, but we cannot distinguish between the two. 

Additionally, assuming that restricting the analysis to people aged 25-45 will give insight into the role of liquidity constraints seems tenuous. There is likely significant heterogeneity in such constraints within this age group. 

Response: 

Thank you for this comment. We have reconsidered it and decided to drop this from our robustness and sensitivity analyses. Instead, we test the robustness and sensitivity of our results by focusing on different alternative measurements for our variables of interest (see page 13-14 of the manuscript).

Can you do another sensitivity analysis looking at when the man is unemployed >6months?

Response: 

The reason why we initially planned to test the sensitivity of the results by focusing on unemployment spells that last between three and six months is because research (Cardona-Sosa et al., 2018) suggests that an increase in the labour supply of women is most likely for unemployment spells of this duration, while for longer unemployment spells households might adapt in other ways (e.g. reducing expenditures). But we also added the sensitivity check you suggested (see manuscript page 13-14). 

I think it would be good to include a research question.

Response: We have included the research question in the introduction (see manuscript page 3)

There are some language errors that will need correction before publication. E.g., in the abstract, “the” should be removed from “Recent changes in the labour markets”.

Response: 

We have gone through the manuscript carefully and hope that we have addressed all language errors. Nevertheless, we plan to send the final version of the paper with all analyses included to a professional proofreader. We have reserved financial resources for this. 

Reviewer #2: 

The scope of the study is to investigate the potential increase of women labour supply once man become unemployment. The value added of the study is to also consider the mediating role of the policy environment. I think the proposed methodology can partially answer to the research question but I have a few doubts.

1. The dependent variables proposed are of two kinds, one, women move from unemployment to employment, two, women move from part-time to full time. I wonder whether the data allow to also have a more sophisticated measure considering also eventual additional hours (both in terms of women increase, but also of men lost, rather than simply the movement to unemployment). This would allow to calculate a sort of elasticity in terms of hours of work reduction from men and increase for women within a couple (and of course the mediating role of politics). 

Response: 

Thank you for your suggestion. We consider it highly valuable, in particular as an additional measure of women’s labour supply increase. However, we are not able to accommodate it, as the data does not allow it. Although the EU-SILC data includes a question on the number of hours usually worked per week in the main job, this is not available for the monthly retrospective data. One of the contributions of our registered report protocol is that it makes use of the monthly retrospective data, which allows us to investigate the added-worker effect at a more granular level compared to previous studies. At the same time, focusing on the reduction in working hours for the male partner would change the scope of our study. The added-worker effect is a theoretical framework developed based on the unemployment of the male partner, which 1) results in an income loss and 2) increases the time availability of the women, so that she can increase her working hours. Additionally, it would require us to reconsider our policy variables (i.e. net replacement rates), as they are based on the man becoming unemployed. We, however, take your suggestion as an inspiration for further studies. 

2. Is EU-Silc the best dataset for this? What about the labour force survey for looking in more detail at the labour market characteristics of men and women?

Response: 

The Labour Force Survey (LFS) is a valuable dataset. The reason why we did not consider it is because the LFS would only allow us to study couple employment transitions during quarters (between 2 and 4) in the same year. This period might be too short to observe employment transitions. In addition, EU-SILC allows us to follow couples during each month, for up to 4 years.

Reviewer #3: 

This registered report is written in clear and concise language, and centres on an interesting research question. Specifically, the report highlights how the researchers aim to analyse the intervening impact of tax benefit policies on the one hand and child care policies on the other hand, on a spouse’s response to her husbands unemployment. The researchers succeed in highlighting why this question is relevant, and have a sound theoretical basis for their hypotheses. 

However, the report describes a number of odd methodological choices, that lead me to doubt whether the proposed research will succeed in reaching its professed aims. In addition, a number of additional operationalization and methodological choices are not outlined in sufficient detail to assess at this stage whether the research will be feasible and replicable.

- The authors cite the total sample size, in terms of couple-months and couples. However, in order to assess the feasibility of the study, more information is needed: how large are the sample sizes by country, especially when also considering the numbers of “relevant” couples, i.e. couples where the male makes a transition from employment to unemployment within the period studied, per observation period.

Response: We have computed the number of observations by country (see table below). However, based on our correspondence with the Editor, we should not compute the number of “relevant” couples or perform any other computations on the data, as it would violate the principles of preregistration. 

Country NW→E

PT → FT

 No. % No. %

AT 31,512 2.61 51,636 8.77 

BE 24,936 2.07 47,640 8.10

BG 35,796 2.97 3,912 0.66

CY 39,204 3.25 11,280 1.92 

CZ 38,664 3.20 6,816 1.16

DE 5,388 0.45 13,836 2.35

DK 1,416 0.12 2,388 0.41

EE 30,144 2.50 11,160 1.90

EL 118,692 9.83 23,208 3.94

ES | 112,488 9.32 40,968 6.96

FR | 58,404 4.84 85,032 14.45

HR | 36,384 3.01 3,048 0.52

HU | 54,540 4.52 6,588 1.12

IE | 29,664 2.46 18,780 3.19

IT | 151,212 2.46 71,700 12.18 

LT | 20,316 2.46 7,536 1.28

LU | 3 9,600 3.28 43,068 7.32 

LV | 25,872 2.14 5,568 0.95 

MT | 62,376 5.17 12,132 2.06 

PL | 89,004 7.37 19,956 3.39

 PT | 53,856 4.46 12,348 2.10

RO | 56,928 4.72 15,540 2.64

SI | 26,976 2.23 8,136 1.38 

SK | 34,164 2.83 4,668 0.79

UK | 29,724 2.46 61,548 10.46

- Including the transition from studying to employment (for the spouse) seems not in line with the proposed research question. 

Response:

Thank you for this valuable comment. Following it we decided to drop women's transitions from studying to employment and we now consider only transitions from being out of work (inactive, unemployed) to being employed. We have recomputed the sample size based on this specification (see manuscript page 7).

- It is not entirely clear what the observation period is, for how long and which couple employment transitions are included. It may help to extend figure 1 with a number of additional examples, in terms of promoting replicability.

We have extended figure 1 so as to include examples of possible employment transitions of the couples (see manuscript page 10). We have also rewritten the two paragraphs which were placed before the figure in order to make it clearer which transitions are included and how long couples are observed. With regard to the observation period, we also rewrote the paragraphs emphasizing that we begin following couples in the first month in which they are surveyed and we stop following them until the last month they are followed in the survey (tn) or until the man becomes inactive, a woman’s labour supply declines or the union dissolves, whichever comes first. 

- My main issue with the report is the proposed approach to take account of tax benefit policies. The authors propose to derive hypothetical NRRs and MTRs for a large share of hypothetical households using the hypothetical household tool from EUROMOD, in order to assess how the incentives of the tax benefit policies impact on the spouses response to her husband’s unemployment spell. However, the main advantage of using EUROMOD would precisely be to be able to calculate at the individual level the actual NRRs and MTRs faced by each individual, both in the affected households as for the unaffected households. Given that they clearly are aware of the possibilities of EUROMOD, I see no reason why the authors would opt for the less precise operationalization of the impact of the tax benefit system in their approach. I can imagine that using actual incentives might make it less self-evident to use a country-level assessment of the tax benefit policies, but in that case the methodology should be changed, rather than choose a second-best option to fit the model specification. 

Response:

Thank you for offering us the opportunity to further clarify our methodological choices. Indeed, EUROMOD is a tool designed to calculate the NRRs and MTRs for each individual in a household, by making use of EU-SILC data that is specifically reconstructed for this purpose (in order to obtain this data researchers have to follow an official application process which involves submitting a Research Project Proposal, which needs to be approved by Eurostat; for more info see https://euromod-web.jrc.ec.europa.eu/download-euromod). We decided to make use of the hypothetical household tool because it is simply not possible to use our longitudinal EU-SILC data with EUROMOD (we changed the manuscript so as to specifically mention it, see manuscript page 10). This is for two reasons:

First, the EU-SILC data as made available for research purposes is not compatible with EUROMOD. More specifically, the EU-SILC data has a different coding and construction of variables compared to the data format required by EUROMOD (specific information on the data format required by EUROMOD for each country can be found in the document “EUROMOD Input Data 2020 - codebook”, available here: https://euromod-web.jrc.ec.europa.eu/resources/model-documentation). 

Second, adjusting and recoding our longitudinal dataset so as to make it compatible with EUROMOD is also not possible. EUROMOD makes use of cross-sectional data, which includes variables that are not made available in the longitudinal data (e.g. hh070: housing cost; pl073 - pl076: number of months spent at full-time/part-time work as employee/self-employed) and the identification numbers of respondent in the cross-sectional and longitudinal datasets are different (a change which was introduced by Eurostat for anonymisation purposes).

Thus, we were left with two possibilities: a) using the EU-SILC data reconstructed for EUROMOD and computing general year-country-average NRRs and MTRs or b) using the hypothetical household tool (which requires considerably more effort and time from our part) to compute NRRs and MTRs for households that resemble as much as possible the households in our dataset. We decided to opt for the latter option, although not ideal, because by contrast to the former it allows us to capture the differences in NRRs and MTRs based on household composition. 

- The choice to assess availability to child care by looking at coverage rates is defensible. However, as the authors themselves highlight the main reason for this choice is pragmatic, an obvious robustness check would be to include the SILC information on current use of formal and informal child care, especially in the part-time case. Getting access on a relatively short term notice (only 3 months after all) is definitely at odds with the reality in a number of European countries. One could assume that it is easier to extend child care hours, than to claim a new spot altogether on short notice. 

Response:

Thank you for your comment. Unfortunately, the data concerning the use of childcare is not available in the longitudinal EU-SILC data that we are using. As such, we cannot run the robustness test you suggested. For more information on the difference in the variables included in the two datasets please see https://www.gesis.org/en/missy/materials/EU-SILC/documents/guidelines. It is true that getting access to childcare at a short notice is difficult. Still, it will be more difficult in countries where childcare is scarce than in countries with high childcare availability. Furthermore, using an actual childcare use instead of the availability is problematic as well. A woman may be using part-time childcare and may not be able to increase her childcare supply because full-time childcare is simply not available to her. We thus think availability indicators also have merit. 

Finally, we would like to stress that we improved our measures of childcare availability compared to what we included in the first version of the manuscript. Following the comment of the reviewer 1, we decided to use a wider array of childcare indicators, i.e. not only for children aged 0-3 but also for children aged 4 to 6 years of age, and 7 to 12 years of age. We also account for full-time / part-time childcare availability (see manuscript page 10).

Finally, I have a number of smaller remarks regarding the literature review.

- The references cited on the added work effect are relatively dated, or, alternatively, focus on Turkey or Latin-American countries.

Response: In the initial manuscript the distribution our references on the added-worker effect was as follows:

1. Based on the period covered:

a) Older studies (before 2018): 9 studies [(McGinnity (2002); Stephens (2002); Prieto-Rodriguez J, Rodriguez-Gutiérrez (2003); Mattingly MJ, Smith (2010); Hardoy I, Schøne (2014); Karaoglan D, Okten (2015); Gong (2011); Kohara (2010); Ghignoni E, Verashchagina (2016); Addabbo et al (2013);]

b) Recent studies: 8 studies [Bredtmann (2018); Ayhan (2018); Cardona-Sosa (2018); Martinoty (2022); Schøne P, Strøm (2021); Keldenich C, Knabe (2022); Bryan M, Longhi (2018);]

2. Based on the country covered:

a) European countries: 8 studies [McGinnity (2002); Prieto-Rodriguez J, Rodriguez-Gutiérrez (2003); Addabbo et al (2013); Hardoy I, Schøne (2014); Ghignoni E, Verashchagina (2016); Bredtmann (2018); Schøne P, Strøm (2021); Keldenich C, Knabe (2022); Bryan M, Longhi (2018);]

b) Developed countries outside of Europe (USA, Australia, Japan): 4 studies [Stephens (2002); Mattingly MJ, Smith (2010); Gong (2011); Kohara (2010);]

c) Turkey: 2 studies [Karaoglan D, Okten (2015); Ayhan (2018)]

d) Latin-American countries: 2 studies [Cardona-Sosa (2018); Martinoty (2022)]

We would like to emphasize that, to the extent that we cited older studies or studies which did not cover Western countries, we did so because they are highly relevant as background for our study (e.g. McGinnity (2002); Prieto-Rodriguez J, Rodriguez-Gutiérrez (2003) are exceptions of studies that cover multiple countries). Based on your suggestion we have examined the literature and included more recent studies on Western developed countries: Halla (2020); Baldini et al. (2018); Cammeraat et al (2023). 

- The literature review only hints at two very likely determinants of differences in the added worker effect at the household level: cultural preferences, that may differ between countries, but certainly also within countries, by social class or by minority group. The literature review is not explicit about this, and the research also fails to control for these determinants in the proposed methodology. Earnings potential and education is included in the model, but its likely impact is only summarily addressed in the literature review.

Response: We have revised the literature review section so as to more explicitly discuss the role of culture and earnings potential. Following the comment of Reviewer 1, we also included gender-role attitudes in our study (see manuscript page 12). To the best of our knowledge there are no studies that specifically examine the differences in terms of AWE based on social class or by minority group. Some studies control for having a non-western migration background, but do not find a statistically significant association with an increase in the labour supply of women. In our study we cannot assess the role played by social class or by belonging to a minority group, as the EU-SILC data does not include any information on social class, ethnicity or migration status in the longitudinal data

- The increase in men’s employment instability is given as the main rationale for embarking upon the proposed research. However, the reasons for this increased instability leads one to wonder why this would only be a “male” phenomenon. Why would the same drivers have a different impact on female employment? Also, the instability driven by a higher share of fixed employment contracts would not necessarily translate in more mid-term unemployment spells, if they are conducive to a well-functioning labour market. Insiders and outsiders are likely affected differently. 

It was not our intention to frame employment instability as a phenomenon that only affects men. We realized after your comment that the previous version of the introduction might have given this impression. We are grateful to you for drawing our attention to this fact. We have changed the text so as to rectify our mistake (see manuscript page 2-3). 

Reviewer #4: 

The report describes a study which aims to investigate women’s labour market choices when their partners have become unemployed, and whether welfare state policies modify their responses, focusing on childcare and tax/benefit policies. The topic is highly relevant, and the study has a potential to provide valuable new information on partners’ interlinked labour market choices in couples.

The aim and design of the study are adequately described. The protocol provides relevant literature and discusses research gaps, which their study will address. Hypotheses (H1-H5) are clear, and can be investigated with the chosen methodology and data. 

Response: Thank you!

Data (EU-SILC) are sufficient for conducting the planned study. The study aims to use Eurostat information on childcare coverage and EUROMOD/UKMOD simulation tools to provide information on the welfare policies.

I have some suggestions for the authors to consider:

Country-level indicators

The study aims to examine the role of taxation/benefits and childcare policies in potentially modifying women’s responses to their partner’s unemployment. However, I wonder if it is eventually possible to distinguish the role of NRR, MTR, or childcare availability if they go hand in hand, e.g. a country with high childcare availability may also be a country with high MTR (for example, the Nordics). I take that this remains to be tested empirically, but may be worth considering already before analyses, and to avoid overstating the conclusions. 

Response: We agree that policies can simultaneously influence the labour supply increase of women. We decided we will introduce them jointly into the model if we experience no multicollinearity problems. We incorporated this change on page 13.

There could also be other (policy) factors than those examined in the analyses which could explain country-level variation in women’s responses. For example, if (women’s) part-time work is very common in a country, transition from inactivity to (part-time) employment may be easier even with small children. 

Response: We agree that there are other policy factors that can play a role in women’s labour supply increase, the prevalence of part-time work being one of them. In this study we decided to focus on childcare and the design of the tax-benefit policies, which are often mentioned in the literature in the context of AWE but have never been explicitly tested. In the current version of the study we have detailed measures of childcare availability (full-time/part-time coverage for children in three different age groups) and very complex NRR and MTR measures which account for household situations. We think this is the point at which we need to stop and leave other policy variables for future research, otherwise the paper becomes too complex. We do, however, control for some macro-level variables such as unemployment rate or female employment rate which should capture availability of jobs (incl. part-time jobs). 

Analyses of the transitions

It was unclear how the male partner’s return to employment (after a period of unemployment) is treated in the analyses.

Response: We continue observing the couple when he returns to employment following unemployment. We do not treat such an employment spell differently than the other employment spells. Only couples in which the man becomes inactive, a woman’s labour supply declines or the union dissolves are censored, which we explain on page 10. 

Also, how is union dissolution taken into account – are couples censored at the time of dissolution? (and at what precision – monthly, annually – is the time of union dissolution measured?)

Response: In order to build our sample we first identify the men and women who have a partner and in the second step we match them based on the spouse ID. Consequently, in building our sample we only select those couples that are in a union (either cohabitation or marriage) and couples whose union ends are censored following their dissolution. With regards to the precision with which the time of union dissolution, the dataset allows us to identify dissolutions yearly, as the monthly retrospective questions concern exclusively the employment status. 

Calculations of NRR/MTR

The authors treat partners in cohabiting couples as singles when calculating NRR and MTR. I take that single here means single(parent)+children, when cohabiting couples have children, and not a single-person household (?). 

If I understand correctly, treating cohabiting couples differently from married couples is based on how they are treated in national taxation and benefit policies. I wonder, however, if in many countries cohabiting couples resemble married couples (in terms of taxation and many benefits) when they have common children? E.g. in the case of cohabiting couple with children living in the household, would the ‘match’ in EUROMOD/UKMOD be a married couple with children, rather than a male/female single adult (with children)?

Response: Thank you for this observation. We have checked the EUROMOD documentation to identify better ways of treating cohabiting couples. Based on the EUROMOD codebook (see: https://euromod-web.jrc.ec.europa.eu/resources/model-documentation) if the two partners are cohabiting, their marital status should be defined as single. At the same time, if there are countries in which cohabiting couples that live with children are treated similarly to the married couples, EUROMOD would account for this in estimating the household income. We follow this approach in generating the hypothetical household data. The relationship of the members in the hypothetical household is defined through partner ID and mother\\father ID.

I assume that the NRR and MTR strongly depend on personal/household income. For example, MTR varies by income level, and is generally higher at lower income levels (at least in countries which provide various means-tested benefits for families). Using the average EU-SILC wage to calculate disposable income in reference households ignores this aspect, if I understood the procedure correctly (that the authors are assigning an average country-level wage to households with full-time employed male/female partner). If my interpretation is correct, would it be meaningful to calculate MTR in the lowest /two lowest income quintiles (income quintiles calculated from the EU-SILC data)? 

Response: We use the average percentage of the EU-SILC wage because this is the option offered by the hypothetical household tool (HHT). We do not compute it, but it is rather built-in to the setting of the application, and the user can change the percentage (between 0 and 100%). Thus, we cannot complete MTRs for lowest /two lowest income quintiles, as income is not a built -in feature of the HHT. However, your suggestion is highly relevant. We thus set the income of full-time workers as 67% of the EU-SILC wage and part-time workers as 33% of the EU-SILC wage and we test the sensitivity of our results by varying the wage levels, see page 14 of the manuscript. 

Childcare statistics

While statistics on childcare coverage (participation rates) are used to measure childcare availability, they are not optimal as in some countries the participation rates of children aged 0-2-years are low not due to low availability of ECEC services for toddlers, but (partly) due to that parental leaves or extended parental leaves allow parent(s) to stay at home longer (for example, Estonia, Malta, Hungary, Finland, see LeaveNetwork Country reports and Cross country reports on ECEC, https://www.leavenetwork.org/introducing-the-network/). I do not suggest to use other sources, however, as the Eurostat participation rate statistics are superior in the country comparison. 

Response: Based on the comment raised by the Reviewer 1 we have improved the measure of childcare availability. In the current version we not only use the enrolment in childcare by the youngest children (0-3) but we also include childcare coverage for children aged 4 to 6 and 7-12. We also distinguish between the general percentage of children enrolled in formal childcare and the percentage of children enrolled part-time and full-time in childcare. These changes required us to change the data source. We now compute childcare coverage using the cross-sectional EU-SILC. We think, however, that thanks to this change we are able to study the role of childcare institutions more comprehensively. 

Reviewer #5: Dear authors. Thank you for submitting the protocol to the editorial office. I have only two comments about the manuscript.

Please specify why you chose the Linear Probability Model (LPM) for modeling and not the Logistic Model or Probit Model more commonly used in similar situations.

What assumptions does LPM have and does your data meet them?

Response: We decided to opt for a Linear Probability Model (LPM) as opposed to the Logistic Model or Probit Model for very pragmatic reasons. As we also mentioned in the manuscript (see manuscript page 12-13) LPM offers the advantage of producing results that can be interpreted as differences in probabilities, and are thus intuitively meaningful. Although the use of LPM has been criticized because a predicted probability may fall outside the range 0–1, simulations show that the predicted probabilities from linear and logistic regression analyses are nearly identical (Hellevik, 2009), the only difference residing in the ease of interpreting the results of the former. Moreover, when multilevel models are considered, logistic models can suffer from problems of convergence, in which case LPMs are preferred (e.g. Haapanala, 2022). As we are not able to run any analyses on the data at this point in time, we cannot assess the risk of encountering convergence problems or to test whether our data meets the assumptions of the LPM. However, we consider that given the practical advantages LPMs offer, they are far more likely to converge and should not result in biased estimates.

Reviewer #6: Overall, I find the research problem topical and important from both a theoretical and practical point of view. The moderating role of childcare and tax and benefit policies on the labour supply response of women to the unemployment of their partners has not been thoroughly nor empirically studied and research in this area could lead to relevant findings and proposals for policies.

I find the hypotheses to be in line with the general research objective and grounded on sound theoretical foundations. The literature review briefly describes the most recent and relevant findings in the literature and is well-connected to the hypotheses. 

Data and variables are described in small-enough detail for the reader to understand the methodological design of the study. 

Response: Thank you!

However, the equations for the models that will be used to test the different hypotheses should also be provided and explained. 

Response: We have included the equations for the models we plan for each of our hypotheses in the manuscript (see manuscript page 13)

Some details would also be welcome on the reasons for choosing each control variable that is added to the model, supported by other studies using the same or similar variables. 

Response: We have included reasons for the control variables we consider relevant (see page 12).

The authors should provide some information on where the data will be made available when the study will be completed.

Response: The data cannot be made available by us, as we have to follow the rules of EU-SILC data use. However, any researcher can apply individually to access the EU-SILC data to the Eurostat and run our code on this data. We will make the code available upon publication of the article in a data repository. We will also provide links to the website which describes how the data can be accessed. Thus, our analysis will be reproducible. 

Some propositions for robustness checks have been formulated. I suggest the authors to also look at the opportunities offered by the European Commission’s Tax and Benefits Indicators database for additional variables for robustness checks.

Response: Based on the suggestions of the reviewers we have included more robustness checks in our plan (see manuscript 18). We have also examined the European Commission’s Tax and Benefits Indicators as suggested by you. It includes the tax-benefits indicators of interest (NRRs and MTRs) derived using EUROMOD, but less detailed than the indicators we compute using the HHT. For example, the NRRs are available for one earner couple without children or with two children. On the other hand, we compute the NRRs for more diverse family types (see manuscript page 11 and the supplementary information). Consequently, we decided against the use of the Tax and Benefits Indicators, as our approach already provides considerably much more detailed information. 

Last but not least, although I appreciate the overall quality of the report as very high, further attention to some minor details is required. 

For example, I would see the research objectives presented in the introduction, before formulating the research hypotheses. 

There are some inconsistencies in the text regarding the number of countries in the sample (24 EU Member States on p. 8 – nothing about the UK, and 25 European countries on p. 13). Moreover, the period 2009-2019 covers 11 years and not 10, as stated on p. 13. 

Response: In the introduction we include both the research question and the objectives of the paper (see page 3; the last paragraph). 

We have gone through the manuscript carefully and addressed the inconsistencies you mentioned as well as the language errors. 

I congratulate the authors for their research plans and wish them good luck and good results!

Response: Thank you!

Reviewer #7: - In the introduction section you should present the methods used for analysis, the results obtained and also the usefulness of your results

- the results and discussion section is missing. In the section Model specification, you talk about using regression to test the hypothesis, but you do not present the effective results. You should present some tables with the results. Have you effectively analysed the data? You used only regression? This part is not clear. 

- The conclusion section is missing. This section should contain the main results of your paper and the value added generated by your study. Also, you should describe which are the limitations of your paper.

References:

• Giulia Briselli and Libertad Gonzalez Are Men's Attitudes Holding Back Fertility and Women's Careers? Evidence from Europe. Working paper

• Haapanala (2022) Carrots or sticks? A multilevel analysis of active labour market policies and non-standard employment in Europe. Soc Policy Adm.;56:360–377

• Hellevik O. Linear versus logistic regression when the dependent variable is a dichotomy. Qual Quant. 2009;43:59–74.

---

## [Editor Report · Decision Letter 1]

27 Sep 2023

His unemployment, her response, and the moderating role of welfare policies in European countries

PONE-D-23-15097R1

Dear Dr. Matysiak,

We’re pleased to inform you that your manuscript has been judged scientifically suitable for publication and will be formally accepted for publication once it meets all outstanding technical requirements.

Kind regards,

Daphne Nicolitsas

Academic Editor

PLOS ONE
---

## [Editor Report · Acceptance letter]

14 Nov 2023

PONE-D-23-15097R1 

His unemployment, her response, and the moderating role of welfare policies in European countries 

Dear Dr. Matysiak:

I'm pleased to inform you that your manuscript has been deemed suitable for publication in PLOS ONE. Congratulations! Your manuscript is now with our production department. 

Kind regards, 

on behalf of

Dr. Daphne Nicolitsas 

Academic Editor

PLOS ONE